# RIG-I-like receptors direct inflammatory macrophage polarization against West Nile virus infection

Amy E.L. Stone[1,2], Richard Green[1], Courtney Wilkins[1], Emily A. Hemann [1] & Michael Gale Jr. [1]

RIG-I-Like Receptors (RLRs) RIG-I, MDA5, and LGP2, are vital pathogen recognition receptors in the defense against RNA viruses. West Nile Virus (WNV) infections continue to grow in the US. Here, we use a systems biology approach to define the contributions of each RLR in the innate immune response to WNV. Genome-wide RNAseq and bioinformatics analyses of macrophages from mice lacking either RLR reveal that the RLRs drive distinct immune gene activation and response polarization to mediate an M1/inflammatory signature while suppressing the M2/wound healing phenotype. While LGP2 functions to modulate inflammatory signaling, RIG-I and MDA5 together are essential for M1 macrophage polarization in vivo and the control of WNV infection through potential downstream control of ATF4 and SMAD4 to regulate target gene expression for cell polarization. These analyses reveal the RLR-driven signature of macrophage polarization, innate immune protection, and immune programming against WNV infection.

[1] Department of Immunology and Center for Innate Immunity and Immune Disease, University of Washington, Seattle, WA 98109, USA. [2] Department of Basic Sciences, Touro University Nevada, Henderson, NV 89014, USA. Correspondence and requests for materials should be addressed to M.G.Jr. (email: mgale@uw.edu)

RIG-I-like receptors (RLRs) are cytosolic pathogen recognition receptors (PRRs) that detect RNA virus infection. The RLR family consists of RIG-I, MDA5, and LGP2. RIG-I and MDA5 are signaling effectors[1] while LGP2 is a regulator of RLR signaling[2,3]. During virus infection, RLRs bind to viral RNA and activate the adaptor protein, MAVS leading to the induction of inflammatory and antiviral genes, including interferons (IFN) and IFN-stimulated genes (ISGs). RLR signaling also directs the regulation of additional transcription factors to impart innate immune activation and immune polarization, but the RLR- transcriptome and responsive factors that mediate these processes are not defined.

RLRs are critical for immune defense against a range of RNA viruses[4] including flaviviruses that are recognized through both RIG-I and MDA5[5], yet the specific contributions of each receptor to the innate immune response are unclear. West Nile Virus (WNV) is a neurotropic flavivirus that is the most common arboviral infection and a major cause of viral encephalitis in the USA[6]. With no treatments or vaccines available, WNV represents a public health concern[7]. Innate immunity is essential for controlling WNV wherein RLR signaling induces innate immune defenses that restrict and control infection and viral neuroinvasion from peripheral sites[8]. RLR signaling serves to regulate the character of the acquired immune response against WNV[8] and is essential for systemic control of infection and pathogenesis[9]. Importantly, macrophages are pivotal for control of WNV infection in most tissues. They are a major tropic cell of WNV and serve to control WNV neuroinvasion[10]. However, the role of RLRs in programming the macrophage response to WNV infection is unknown.

In mouse models of WNV infection, subcutaneous virus challenge first infects lymphoid resident macrophages followed by infection of splenic macrophages[11] before migrating to the CNS[12,13]. Macrophages are functionally categorized into two broad groups: proinflammatory function (M1) or wound healing anti-inflammatory actions (M2), with each linked with specific functions[14]. During virus infection, externally-derived signals such as cytokine signaling or pathogen recognition drive macrophage polarization to M1 or M2 phenotype. M1s recruit immune cells to the site of infection while M2s resolve the inflammatory response[14]. M1 macrophages arise as part of the immune response to infection mediated by the transcription factors NF-κB, IRF3, and STAT1[15]. M2 macrophages take on tissue repair and resolution of inflammation through activation of the transcription factors STAT3 and STAT6[15]. The induction of one phenotype commits the macrophage to a single polarization until the microenvironment changes[14]. However, the role of the RLRs in this process are not known.

Here we apply a systems biology approach employing transcriptomic and functional analyses of primary macrophages to assess RLR regulation during WNV infection. Our study shows that each RLR plays a unique role in protection against WNV. We define the RLR transcriptome and reveal that RLR signaling drives an M1 phenotype to control WNV infection. Our results show that RLR signaling links to ATF4 and SMAD4 transcription factors and regulates M1 and M2 gene expression that direct macrophage polarization to impart innate immunity against WNV.

## Results

### RIG-I-like receptors control West Nile virus infection. Independent studies have revealed roles for RIG-I, MDA5, and LGP2 in the control of WNV infection[16]. We directly compared the susceptibility of mouse cohorts lacking the RLRs alone or in combination. During WNV infection, mice lacking each RLR had distinct infection outcome (Fig. 1a). Comparative analysis shows that WT mice are overall less susceptible (71.8% survival) than any RLR knockout (range 50–0% survival). Loss of both RIG-I and MDA5 phenocopied mice lacking MAVS, as all mice

succumbed to infection within 8 days after viral challenge. While mice lacking RIG-I or MDA5 had similar increased WNV susceptibility over WT, LGP2$^{-/-}$ mice succumbed to WNV at a higher rate, likely reflecting specific functions in effector T-cell expansion[17]. These observations indicate that each RLR plays a unique role in directing the outcome of WNV infection in vivo where one RLR does not compensate for lack of another.

We utilized our cohort of RLR knockout mice to assess RLR-dependent responses in WNV infection of primary bone marrow-derived macrophages (BMMs). BMMs were derived from WT, MDA5$^{-/-}$, LGP2$^{-/-}$, RIG-I$^{-/-}$xMDA5$^{-/-}$ DKO (DKO), or RIG-I$^{-/-}$ mice. These BMMs expressed equivalent levels of each RLR, except the targeted knockout (Fig. 1b). As Toll-like receptor (TLR)3 and 7 have been implicated as PRRs of WNV[16], we determined the levels of these TLR to be comparable in WT and DKO BMMs (Fig. 1c) and functionally active (Supplementary Fig. 1). WT and DKO BMMs induced similar amounts of IFNβ in response to TLR stimulation. Notably, when infected with WNV, WT cells, but not DKO cells, induced a high level of IFNβ expression despite the presence of functional TLR3 and TLR7 in the DKO BMMs (Supplementary Fig. 1). We found that each BMM genotype supported productive infection but WNV replicated to the highest levels in DKO cells (Fig. 1d). These results demonstrate that RIG-I and MDA5, not TLR3/7, serve as the primary PRRs to direct innate immunity against WNV infection in macrophages.

### RLR-dependent transcriptional programs in macrophages. We performed RNAseq on WT and RLR-deficient BMMs infected with the WNV-TX02 infectious clone[18] and analyzed the results through an immuno-informatics pipeline (Fig. 2a). We compared global gene expression analysis of mock to WNV infection of each separate genotype to identify differentially expressed (DE) genes (Supplementary Data 1). The BMMs had similar numbers of DE genes except DKO samples, which had significantly fewer DE genes despite having a similar number of overall reads (Fig. 2b). DKO BMMs had the highest percent of viral reads (Supplementary Fig. 2A), with viral genome sequence reads distributed uniformly across the WNV genome (Supplementary Fig. 2B). As the backgrounds of the mice varied, we compared the B6 mock samples to the RIG-I$^{+/+}$ mock samples and found that the WT samples were highly similar (Supplementary Fig. 3, r2 = 0.975). We used the B6 WT as our representative WT in future analyses.

Figure 2c shows a heatmap of the DE genes ($p > 0.01$, 2-fold minimum change) of each genotype compared against WT DE genes. Genes were grouped based on similar expression patterns to reveal modules of functional categories. We defined five major gene modules, with innate immunity and immune-regulatory genes dominating the response modules. These analyses also revealed that WNV infection suppresses extracellular matrix (ECM) organization genes, which is characteristic of activated macrophages increasing chemotaxis and cell movement[19]. Comparing our DE genes to the Matrisome[20], we found that WNV-infected BMMs were actively changing their ECM genes in an RLR-dependent manner (Supplementary Fig. 4). Global gene expression examination revealed striking differences between the single knockout cohorts and the DKO cells. Most DE genes identified in the WT response failed to change in the DKO BMMs during WNV infection and some genes had opposite directionality. These results show that DKO cells fail to initiate a specific transcriptional response against WNV infection despite having functional TLRs. The DE deficits in DKO cells thus represent RLR-dependent genes that contribute to activation of macrophages for viral control of WNV.

Linking the DE genes to immunity among the genotypes, we found the GO classifications of immune response (Supplementary Fig. 5A, Supplementary Data 2) or innate immune response

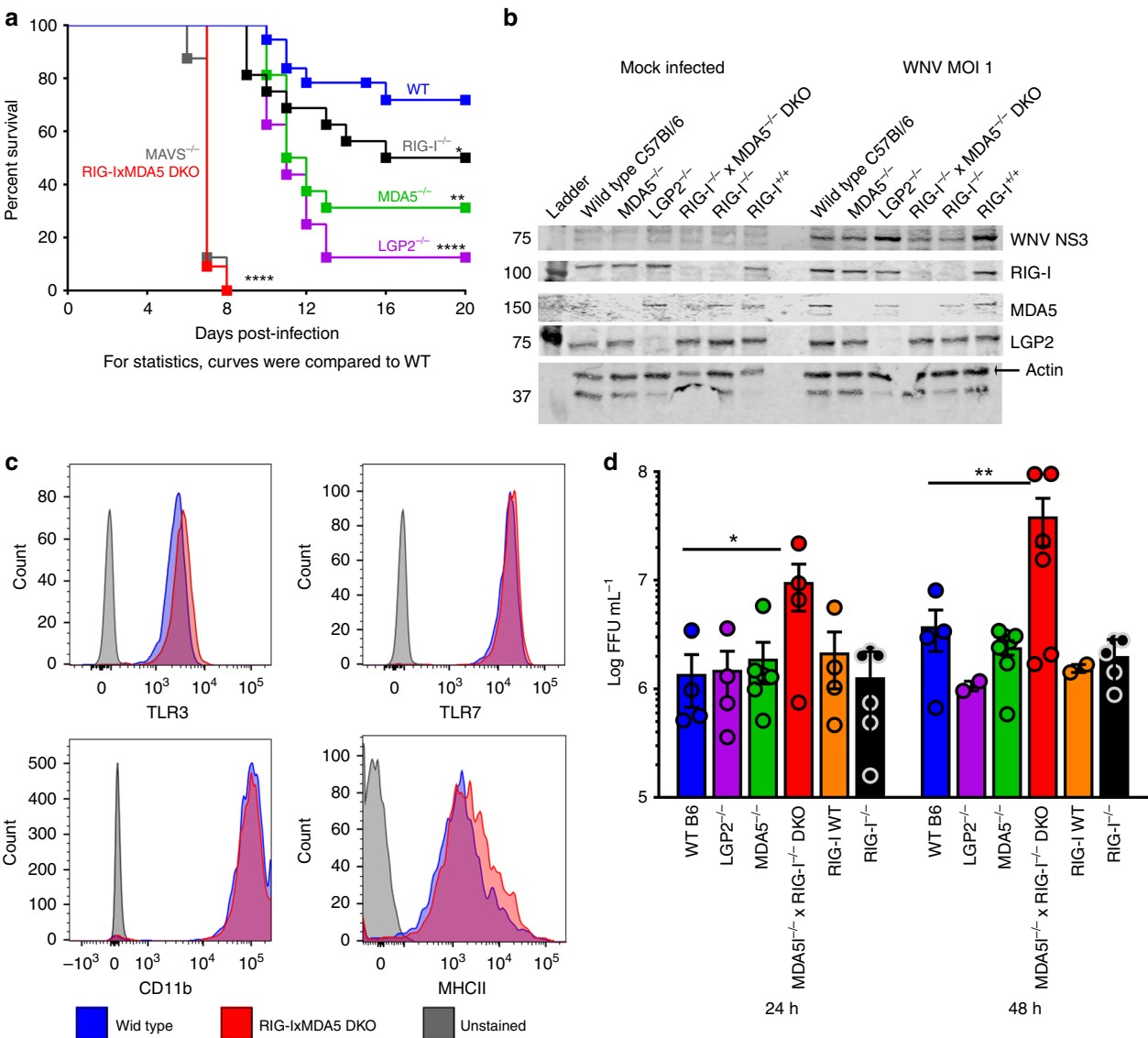

**Fig. 1** RIG-I-like Receptors (RLRs) impart control of WNV infection. **a** Mice were infected with 100 pfu WNV in the footpad and monitored for survival for 21 days. Percent survival is shown. Significance stars for each line compared to the Wild-type (WT) line (combined C57Bl/6 [$n = 24$] and RIG-I$^{+/+}$ WTs [$n = 12$] 36 total mice) are shown. RIG-I$^{-/-}$, MDA5$^{-/-}$, and LGP2$^{-/-}$ $n = 15$ mice each; MAVS$^{-/-}$ $n = 7$ mice; RIG-I/MDA5 DKO $n = 10$ mice. **b** BMMs from each genotype were infected or mock infected with WNV at MOI 1 and harvested at 24 h post-infection. Cell lysates were assayed for the presence of RLR proteins, WNV NS3 protein, and actin by western blot. **c** WT (blue) or DKO (red) BMMs were stained for flow cytometry for TLR proteins or macrophage markers. Histograms for each assayed marker are shown. **d** BMMs from each genotype were infected for 24 or 48 h with WNV at MOI 2.5. The supernatants were harvested and assayed for viral load by focus forming unit (FFU) assay. Bars represent the mean viral load ± SEM from $n = 3$ independent sets of supernatants assayed for virus in triplicate. Panels **b** and **c** show representative data from $n = 3$ independent experiments. *$p < 0.05$, **$p < 0.01$, ***$p < 0.001$, ****$p < 0.0001$ (log-rank;Mantel-Cox for survival curves and Ordinary one-way ANOVA with Holm-Sidak's multiple-comparisons post-test for ffu analysis). Source data are provided as a Source Data file

(Supplementary Fig. 5B, Supplementary Data 3), severely dysregulated in DKO cells. Examining the IFN response to WNV, the DE genes were divided into ISGs (as defined by the WT BMM response to IFNβ treatment; Supplementary Data 4) and non-ISGs (Supplementary Fig. 5C/Supplementary Data 5, and Supplementary Fig. 5D/ Supplementary Data 6, respectively). WNV infection induced a strong ISG response but this response was largely absent in DKO cells. ISGs segregate into functional categories consistent with macrophage activation (Supplementary Fig. 5C). By subtracting ISGs (Supplementary Data 4) from the DE set, we identified WNV-induced non-ISG responses occurring in all genotypes except the DKO, thus defining the virus-induced macrophage response to infection beyond the canonical IFN-stimulated response.

**RLR-dependent virus-induced genes**. To dissect the roles RIG-I and MDA5 play in response to WNV, we filtered the overall gene list to identify RLR-dependent genes. Starting from the WT DE list we removed any DE gene that was also found in the DKO cells, leaving only the genes whose differentially expression relies on the presence of RIG-I and MDA5 during WNV infection (Fig. 3a). Using the subsequent heatmap of those genes, we found that many of the RLR-dependent genes are present in the single RLR KO cells, suggesting that compensation between the RLR family members can occur at the gene expression level (Fig. 3b, Supplementary Data 7). While the DKO cells were deficient in regulating ISGs (Supplementary Fig. 6A), they also fail to regulate novel virus-induced non-ISGs found in the WT response (Supplementary Fig. 6B). These analyses suggest that

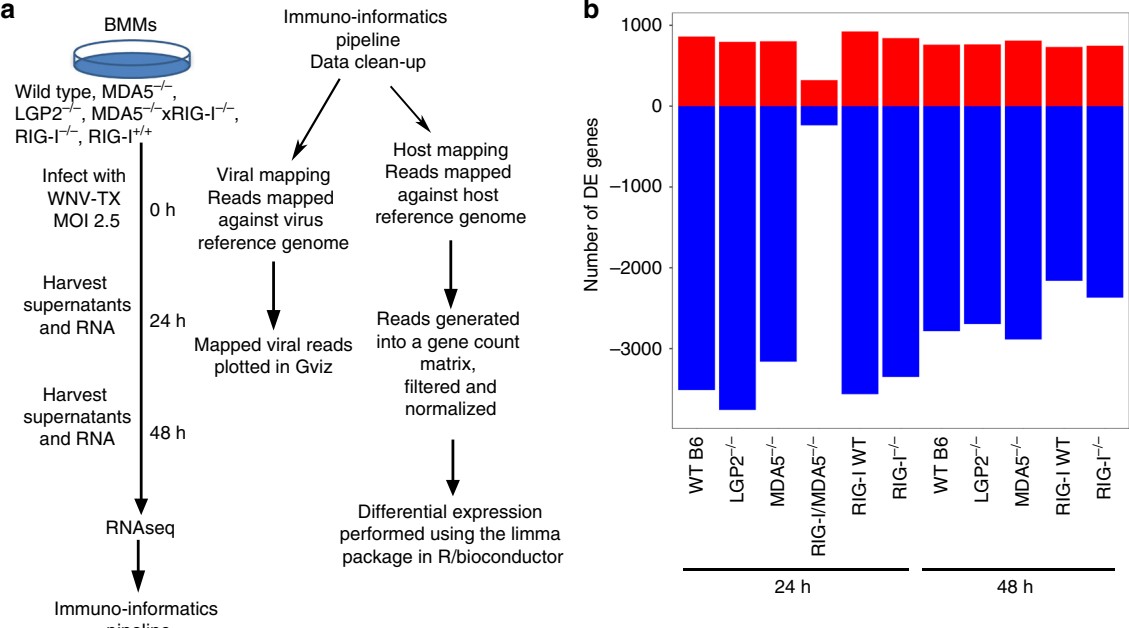

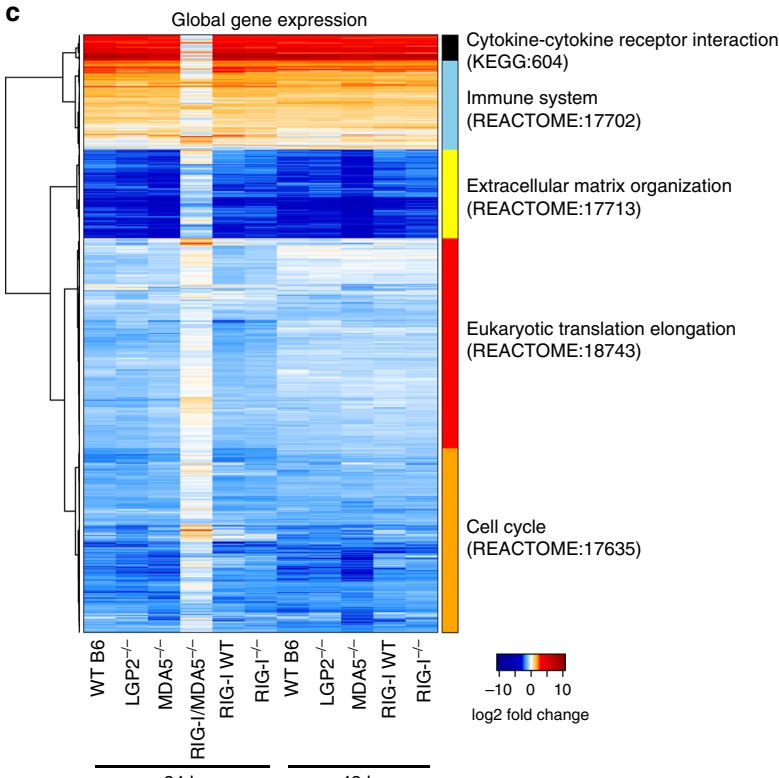

**Fig. 2** RLRs dictate the transcriptome response to WNV infection. **a** Experimental design. BMMs were infected with WNV at MOI 2.5 for 24 or 48 h when RNA and supernatants were harvested for analysis. RNA was submitted to RNA sequencing and analyzed through an Immuno-informatics pipeline. Gene expression was determined by comparing the number of reads in the infected condition compared to that genotype's mock condition. **b** Bar graph representing the total number of differentially expressed genes by genotype and time point. Induced genes are shown in red as positive numbers while suppressed genes are shown in blue as negative numbers. A single value is shown for each genotype and time point. **c**) Heatmap of global gene expression. Genes significantly changed in the WT response compared to the WT mock are shown on the heatmap as horizontal rows. The columns are each genotype at the noted time points. Genes that had more mapped reads in the infected condition are shown in red while genes that had fewer mapped reads in the infected condition are shown in blue. On the right, genes with similar expression are grouped into blocks and assigned functional terms. *n* = 3 independent infections, RNA preparations, and sequencing results. Source data are provided as a Source Data file

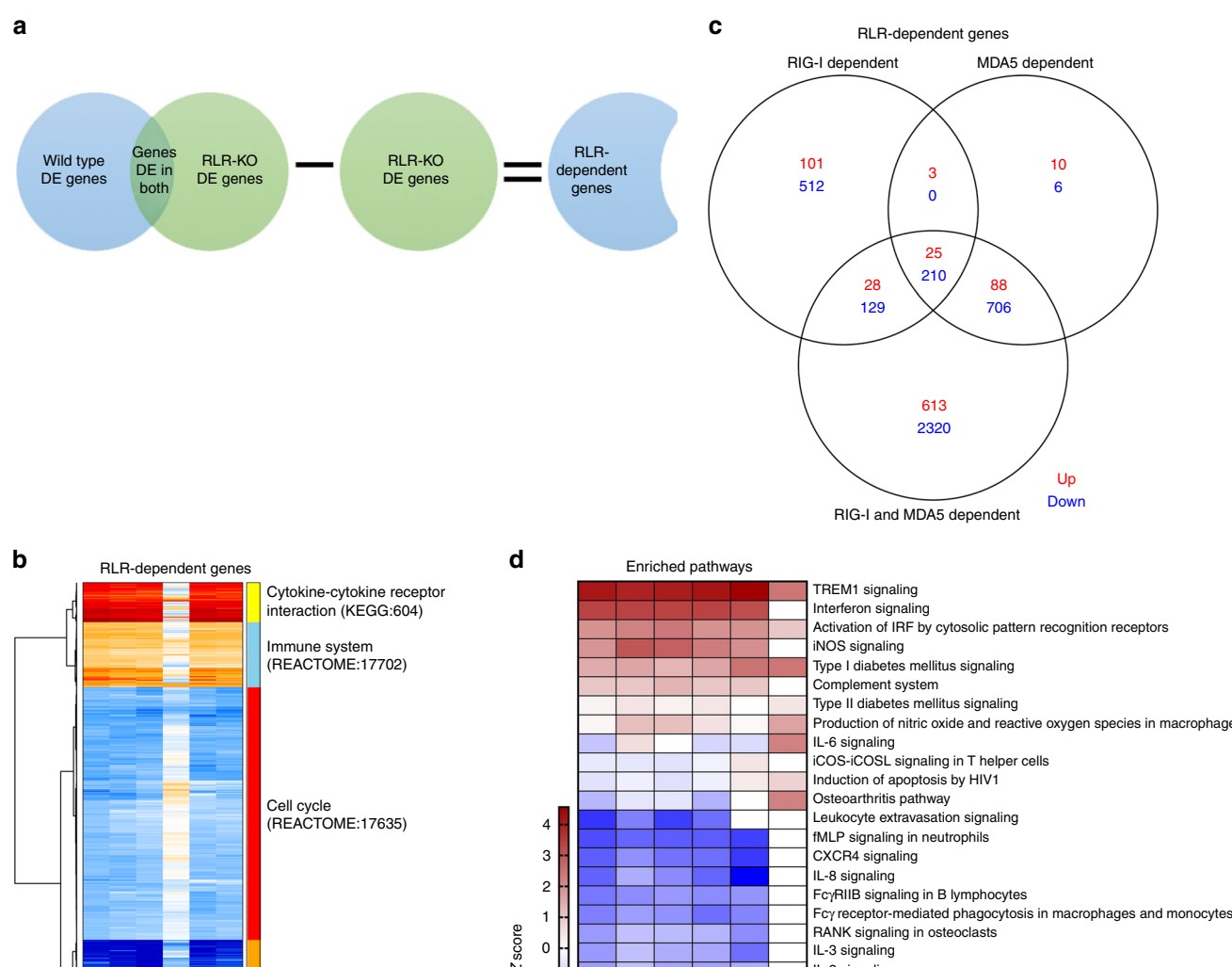

**Fig. 3** RIG-I and MDA5-dependent genes. **a** Schematic of how RLR-dependent genes were determined. **b** Genes that were determined to be dependent on RIG-I and MDA5 (DKO) are shown by heatmap for the 24h time point. Heatmap is set up the same as in Fig. 2b. **c** Venn diagram of the number of genes missing from DKO, RIG-I$^{-/-}$, and MDA5$^{-/-}$ sequencing results compared to the WT response. **d** Pathway analysis (Z-scores) of each RLR-KO DE gene list are shown by heatmap. Red shows pathways that activated while blue shows pathways that are inhibited. Pathways are named on the right. $n = 3$ (DKO), 4 (LGP2$^{-/-}$, MDA5$^{-/-}$, RIG-I WT, RIG-I$^{-/-}$), or 5 (WT B6) independent infections, RNA preparations, and sequencing results. Source data are provided as a Source Data file

the RLRs regulate additional intracellular response pathways beyond IFN to contribute directly to the depth and breadth of the antiviral response. The overlap between RIG-I, MDA5, and RIG-I-&-MDA5-(RLR)-dependent genes among each BMM genotype are shown by Venn diagram (Fig. 3c).

Pathway analysis on the total DE genes from each genotype defined the functional outcomes of RLR-dependent genes (Fig. 3d). WT BMMs induced innate antiviral immunity and inflammatory signaling pathways, and suppressed macrophage chemotaxis and immune regulation pathways, as found in the M1 phenotype[15]. RLR-dependent module expression was altered in the DKO cells, marked by loss of inflammatory signaling modules, more typical

patterns of M2 polarization[21] but with enhancement of inflammatory and apoptotic signaling. These data suggest that RLR-dependent genes contribute to macrophage polarization.

**Transcriptional macrophage polarization requires the RLRs.** To understand how the RLR-dependent transcriptome contributed to macrophage function, we focused our analysis on six functional categories of genes: ISGs, Innate Immune genes, Th1 genes, Th2 genes, M1 genes, and M2 genes. Innate Immune, Th1, and Th2 genes were identified through GO Biological Process lists. M1 and M2 were defined as in a recent macrophage

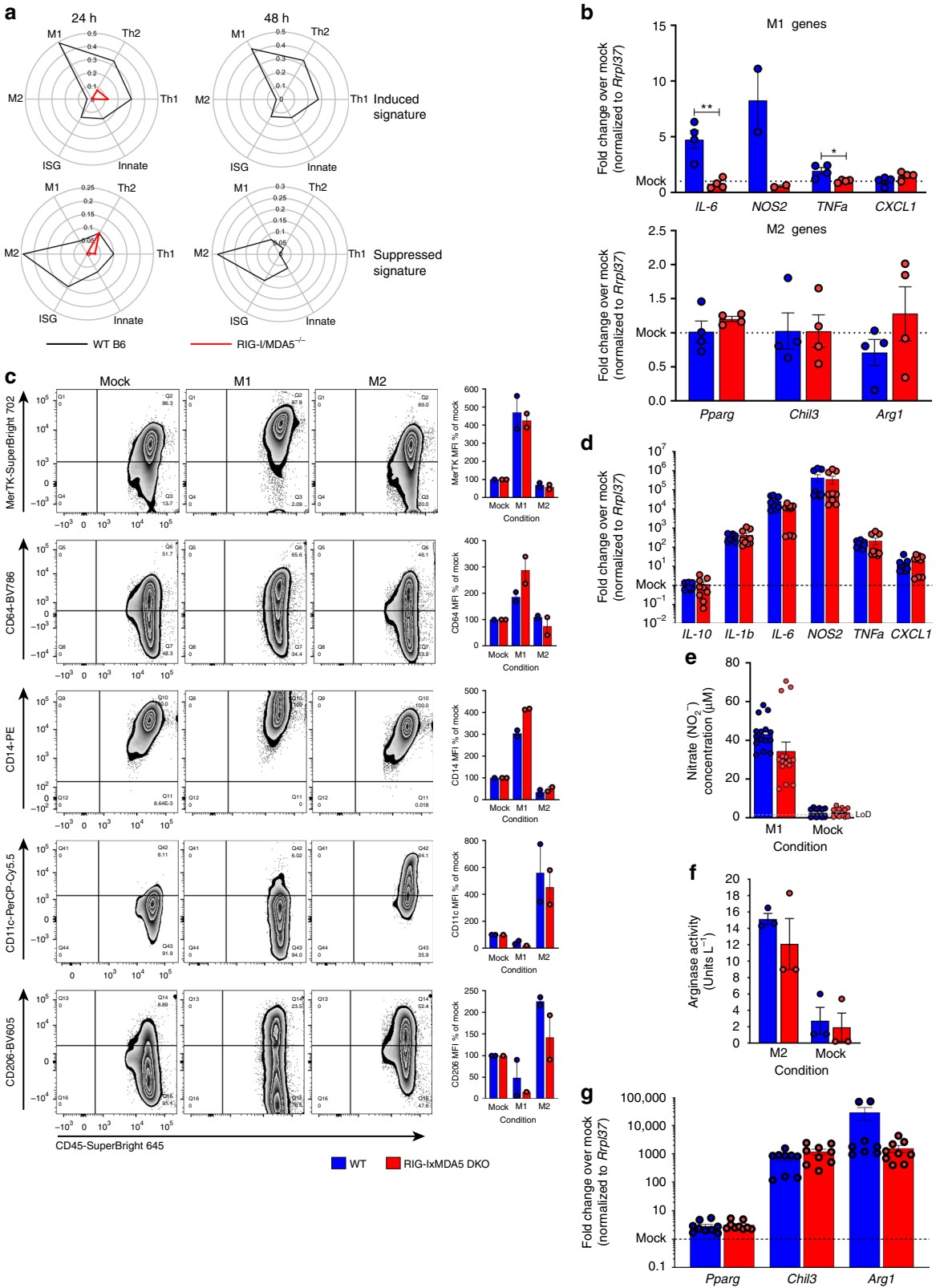

polarization meta-analysis[22]. ISGs were defined by us as described in Supplementary Fig. 5C. The proportion of genes that were significantly induced (or, separately, suppressed) in the WT response for each category out of the total possible genes in that category was determined. The resulting proportions were graphed on a radar plot (Fig. 4a). We found that infected WT cells were dominated by an M1 response while displaying minimal M2 polarization (Fig. 4a, black lines). WT cells also displayed a balanced Th1 and Th2 transcriptional response, while ISG induction and innate immune gene induction were comparably smaller components of the WT BMM response to WNV infection. WT BMMs actively suppress M2 transcriptional responses.

**Fig. 4** WT BMMs polarize to an M1 (inflammatory) phenotype in an RLR-dependent manner. **a** Radar plots of the WT and RIG-I/MDA5 DKO response characterization. The DE genes from the WT samples were separated by time point and whether the gene was induced or suppressed, and then placed into the six defined categories shown. The proportion of genes that were differentially expressed in the WT response out of the total possible genes in each category was calculated. Those proportions are plotted on the spokes, where the higher proportions are plotted further from the center with the WT response connected by black lines. The top radar plots are induced genes while the bottom radar plots are suppressed genes at 24 (left) and 48 (right) hours. The red line overlay is the DKO DE gene proportions for each category. $n = 3$ (DKO), 4 (LGP2$^{-/-}$, MDA5$^{-/-}$, RIG-I WT, RIG-I$^{-/-}$), or 5 (WT B6) independent sequencing results. **b** WT and RIG-I/MDA5 DKO BMMs were infected with WNV at MOI 2.5 for 24 h and then analyzed for M1 (top) and M2 (bottom) gene expression. **c** Flow cytometry characterization of in vitro polarized BMMs. Plots show representative flow plots for polarized WT BMMs. Graphs show the normalized MFI for each of the markers in WT and RIG-I/MDA5 DKO BMMs following polarization. **d** M1 gene expression was analyzed in WT and RIG-I/MDA5 DKO BMMs following in vitro polarization. **e**, **f** Secreted nitrate (**e**) and arginase activity (**f**) were measured from in vitro polarized WT and RIG-I/MDA5 BMMs. **g** M2 gene expression was analyzed in WT and RIG-I/MDA5 DKO BMMs following in vitro polarization. In panels **b**, **e**, **f**, bars are means ± SEM. In panels **c**, **d**, **g**, bars are the mean fold change over the mock condition for each genotype and gene ($\Delta\Delta CT$, normalized to *Rrpl37*) ± SEM. Combined data from $n = 2$ (**b**, **c**, **f**), 3 (**d**, **g**), or 4 (**e**) independent experiments. *$p < 0.05$, **$p < 0.01$ (two-tailed unpaired Holm-Sidak method; modified *t*-tests). Source data are provided as a Source Data file

Together this analysis shows that WNV infection of WT BMMs induces a M1/proinflammatory transcriptome while suppressing the M2/wound healing macrophage phenotype.

To determine RLR-dependence for directing macrophage polarization in WNV infection, we conducted gene expression proportion analysis on the DKO DE genes and overlaid the proportions on the WT radar plots (Fig. 4a, red lines). These analyses show that loss of RIG-I and MDA5 results in loss of virus-induced macrophage polarization. Heatmaps showing the fold change of the M1, M2, Th1, and Th2 genes used in these analyses as well as the overlap of genes among the categories are shown in Supplementary Fig. 7. Representative M1 and M2 genes were verified by RT-PCR following WNV infection in freshly generated BMMs (Fig. 4b). Remarkably, differential gene expression of each functional category is largely absent in the DKO cells (Supplementary Datas 8–11). Taken together, these data reveal that macrophage transcriptional polarization programming depends on RIG-I and MDA5 during WNV infection.

Macrophages can be polarized in vitro using pathogen and host-derived signals that bypass RLR signaling. To determine if RLR-deficiency disrupted macrophage polarization through classic stimuli, we treated WT and DKO BMMs with IFNγ/LPS or IL-4/IL-13 to induce an M1 or M2 phenotype, respectively, and assessed downstream macrophage characteristics. Assessment of cell surface macrophage markers, gene expression, and macrophage functions of nitrite production and arginase activity (Fig. 4c–g), showed that in response to these classical macrophage polarization stimuli, both DKO BMMs and WT BMMs were equivalently polarized to M1 or M2 outcome, respectively. Thus, RLR-deficient BMMs are capable of polarization when activated through classical pathways bypassing RLR signaling but WNV-induced polarization requires the RLRs.

**RLR-specific gene signature and function.** The contribution of individual RLRs to drive gene expression signatures within macrophages during WNV infection was examined, allowing us to identify RLR-specific gene response networks (Supplementary Data 7). We identified genes whose expression is regulated by multiple RLRs, as assessed by the Venn diagram (Fig. 5a). GO analyses identified the functional categories of the RLR-dependent genes that were variably induced and suppressed in expression in a manner dependent on the different RLRs (Fig. 5b). In particular, beyond innate immune genes, we found that RIG-I and MDA5 are essential for expression of gene networks involved in cell metabolism, redox pathways, nucleotide biosynthesis, and translation, all known process that impact macrophage polarization[23]. Overall these results reveal a pattern of gene expression consistent with the progressive distribution of WNV PAMP sensing and signaling by RLRs[5].

We examined the six functional gene categories defined in figure 4 and found that the polarization of BMMs to M1 gene expression with M2 gene suppression was preserved in the individual lines lacking either RIG-I, MDA5, or LGP2 (Fig. 5c). Differences in the magnitude of DE gene regulation in each category were observed, reflecting unique patterns of expression directed by each RLR. When compared with the Fig. 4 data, these results reveal that while any single RLR contributes to the macrophage polarization programming during WNV infection, it is the combination of RIG-I and MDA5 that direct the macrophage phenotype.

**Cytokine production in RLR-dependent macrophage polarization.** RLR-dependent macrophage immune effector function was evaluated by cytokine and chemokine production (Supplementary Fig. 8). WT cells secreted a range of cytokines and chemokines of M1 including CXCL10 (IP-10), MIP-1α, MIP-1β, and IL-6. The overall cytokine levels were similar across the individual RLR knockouts while DKO cells were deficient in cytokine production, revealing a broad impact of RLR signaling in the expression of inflammatory (CXCL5; IL-6) and immune-activating (IL-12) cytokines relevant to the M1 phenotype. These results show that, as with the overall gene signature, both RIG-I and MDA5 are required for inflammatory and immune-regulatory cytokine production during WNV infection.

**Macrophage polarization in vivo during WNV infection.** To determine how RLR signaling directs macrophage polarization during WNV infection in vivo, we assessed macrophage polarization phenotype during WNV infection. Flow cytometry cell analysis to evaluate the activation/phosphorylation state of STAT1 and STAT6, key transcription factors for M1 and M2, respectively, revealed that RLR-deficient mice had dysregulated macrophage polarization in vivo (Fig. 6a, gating strategy Supplementary Fig. 9). Our analyses may include a small proportion of non-macrophage myeloid cells, though, on average, 90+ % of the cells were also CD64+ and MerTK+[24] (Supplementary Fig. 10). DKO mice had a trend of increased absolute numbers of CD11b+F4/80+ macrophages under mock infection conditions. During WNV infection the number of M1 macrophages in the peripheral tissues increased in WT mice but M1 macrophage numbers failed to increase in DKO mice. Yet, M2 macrophages increased in DKO mice (Fig. 6b). The frequency of M1 macrophages also was decreased in DKO mice compared to WT mice, and they failed to significantly increase in response to WNV infection (Fig. 6c). Overall the percentages of M2 macrophages in each tissue were similar between WT and DKO despite the observed reductions in M1 cell percentage. Of note, the DKO mice significantly increased the percentage of M1

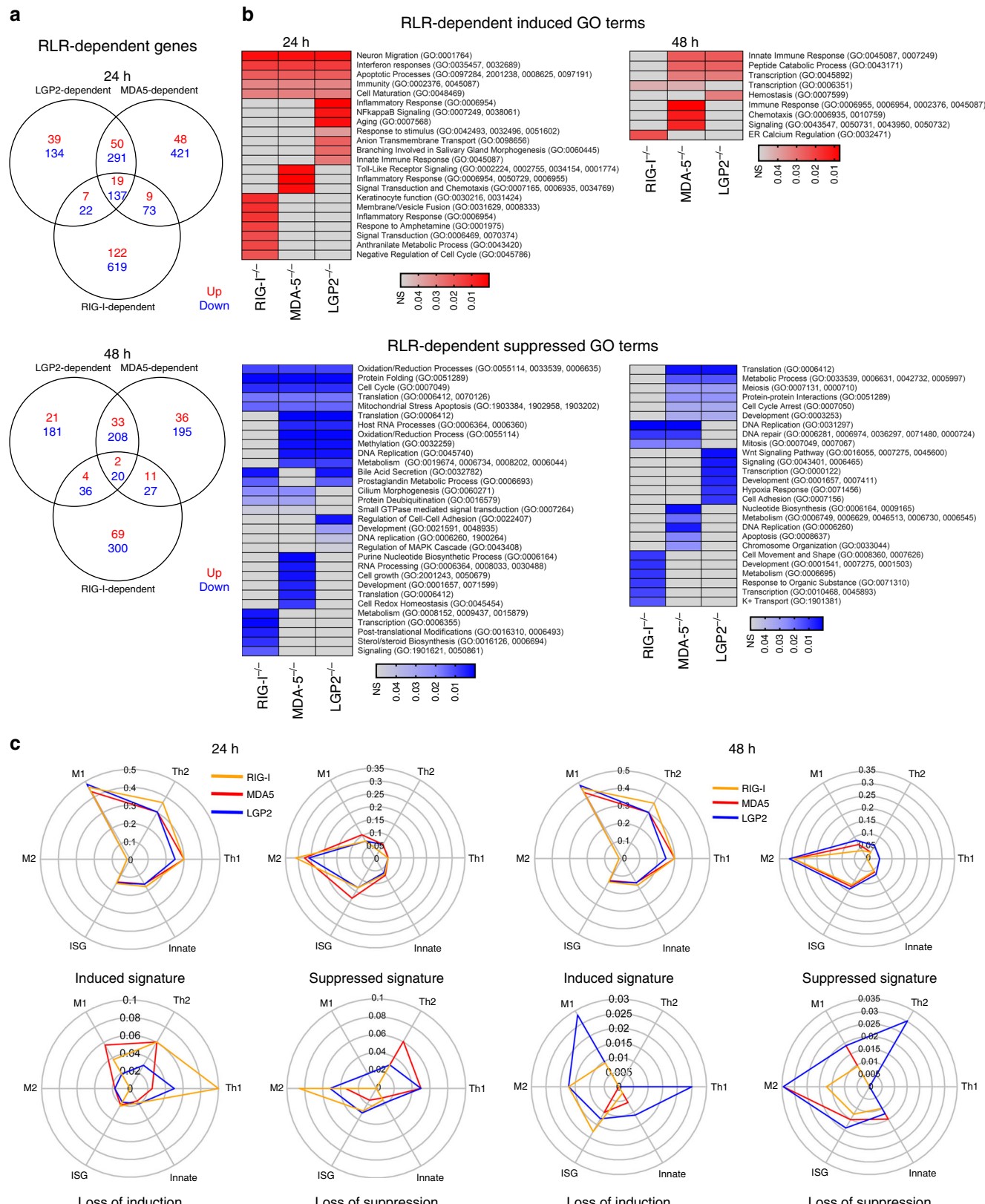

macrophages in the brain albeit with reduced levels compared to WT. We observed more infectious virus production from brain and spleen tissue of DKO mice (Fig. 6d). When BMM were treated with M1 or M2-polarizing stimuli ex vivo we found that M1 macrophage polarization effectively controlled

WNV better than M2 or Mock polarized BMMs regardless of genotype, as shown by FFU and WNV genomic copy number PCR (Fig. 6e, f). These results reveal that the RLR signaling through RIG-I and MDA5 contributes to macrophage polarization in vivo and a component of macrophage differentiation

**Fig. 5** Unique RIG-I, MDA5, and LGP2 transcriptional responses following WNV infection. **a** Venn diagram showing the overlap in numbers of RLR-dependent. Red numbers are induced genes while blue numbers are suppressed genes. **b** Heatmap showing the significantly enriched induced (top, in red) and suppressed (bottom, in blue) GO terms across RIG-I, MDA5, and LGP2 deficient cells. Grey indicates that GO term was not significantly enriched in the samples and are therefore dependent on the specific RLRs for gene module regulation. Each horizontal line represents a GO term while the columns are the genotype of the samples. The color represents the adjusted p-value (co-expression) of the enrichment where a darker color indicates a smaller (more significant) p-value. **c** Radar plots of the RIG-I, MDA5 and LGP2 deficient responses. Genes were categorized, and proportions calculated as in Fig. 4. Top radar plots are the categorization of the genes that were present in each genotype's response. Bottom radar plots are the categorization for the RLR-dependent genes (genes altered in WT response, but unaltered in RLR-KO response). RIG-I is shown in yellow, MDA5 is shown in red, and LGP2 is shown in blue. $n = 3$ (DKO), 4 (LGP2$^{-/-}$, MDA5$^{-/-}$, RIG-I WT, RIG-I$^{-/-}$), or 5 (WT B6) independent infections, RNA preparations, and sequencing results. Source data are provided as a Source Data file

to the M1 phenotype in peripheral tissues for the control of WNV infection in both the spleen and brain.

**ATF4 and SMAD4 may link the RLRs to macrophage polarization**. To assess the linkage of RLR signaling to downstream macrophage transcriptional signatures and polarization phenotypes, we conducted an unbiased bioinformatics analysis to predict upstream transcriptional regulators of the WNV-induced transcriptome (Fig. 7a). This analysis identified two transcription factors (TFs), ATF4 and SMAD4, whose target genes were highly enriched in WT BMMs but were absent in DKO BMMs, indicating that ATF4 and SMAD4 potentially operate as RLR-response TFs. Both TFs are predicted to be inhibited in the WT response but activated in the DKO response. Neither of these TFs has been previously linked to antiviral responses. ATF4 is a central responder to cellular stress, including inducing monocyte/macrophage recruiting chemokines[25,26] while SMAD4 acts in transforming growth factor (TGF)-β signaling and BAFF induction in macrophages[27,28]. ATF4 and SMAD4 each associated with suppression of target genes in the WT BMM response to WNV while suppression is lost in the DKO BMM response to infection (Fig. 7b, c, compare left and right panels, respectively). We found that target genes of each TF included a variety of M1 and/or M2 genes whose expression was found to be differentially regulated in WT and DKO cells. This computational analysis suggests that RLR signaling might link to ATF4 and SMAD4 for macrophage programming following virus sensing.

**Discussion**

Here we have defined the RLR-responsive transcriptome in primary BMM from WT and RLR-deficient mice, revealing that RIG-I and MDA5 signaling programs macrophages to an M1 phenotype for immune protection against WNV. Macrophages are targeted by WNV and are critical for immune protection against virus infection[16] and in neuroinvasion[10]. Signaling processes direct macrophage function and polarization toward inflammatory/M1 or anti-inflammatory-wound healing/M2 phenotypes[15]. We show that RLR signaling, by the combined actions of RIG-I and MDA5, direct a predominant M1 phenotype while suppressing the M2 phenotype within the protective response against WNV infection. Our study reveals that RIG-I and MDA5 are essential during WNV infection serving as PRRs for virus sensing and inducing innate immune defenses that control infection. Our systems-based study identifies ATF4 and SMAD4 as RLR-sensitive downstream TFs potentially regulating macrophage polarizing gene expression.

Our direct comparison of each RLR in protection against WNV infection shows that antiviral immune signaling ex vivo is controlled by the combination of RIG-I and MDA5. The transcriptional signature of macrophages programmed through RLR signaling shows that this response integrates multiple gene modules of broad function to polarize macrophages for immune

response regulation. RLR signaling was essential to produce cytokines/chemokines of the M1 response. The catastrophic loss of the WNV-induced transcriptome in the DKO cells demonstrates that the RLRs are key mediators of the macrophage immune response and immune regulation in general.

Each RLR contributes uniquely to defense against WNV infection. RIG-I was essential for inducing innate immune genes comprising the acute host response to infection while MDA5 linked closely with macrophage production of inflammatory and immune-regulatory genes and cytokines. RIG-I and MDA5 in their roles of PRRs of WNV likely function in series to recognize distinct PAMPs produced during acute WNV infection. In this sense, RIG-I is thought to mediate first recognition of WNV to initiate RLR signaling, which is then amplified upon later by distinct PAMP recognition by MDA5[5]. Our results indicate that this process of RLR signaling in series diversifies the host response to include genes involved in immune regulation such as Th1 and Th2 genes while driving robust IFN and ISG expression. Confirmation of these observations in vivo with RLR-deficient macrophages in an intact host are ongoing.

We identified non-ISG virus-regulated genes that are instead regulated by RLR signaling. These virus-induced genes in macrophages include modules that program the immune response following macrophage polarization and include direct IRF target genes, reflecting the role of IRFs as major downstream transcription factors of RLR signaling[1]. While RIG-I signaling initiates macrophage activation, MDA5 signaling expands the macrophage response to cytokine/chemokine production. In the absence of an RLR, RIG-I or MDA5 can partially compensate to impart macrophage activation and M1 polarization but both RLRs are required for immune protection against WNV. It is important to note that splenic macrophages are essential for the control of WNV infection and neuroinvasion in vivo[11]. Beyond induction of gene expression, we show that RLR signaling directs the suppression of specific gene expression in macrophages during WNV infection. We found that suppressed genes were typically linked to macrophage activation and innate immune effector phenotypes related to differential M1/M2 polarization[29].

Our data sets indicate that LGP2 imparts an amplification of gene expression within modules of interferon signaling and cell differentiation/activation relevant to M2 and Th1 suppression/Th2 activation. LGP2 has been defined as a regulator of RLR signaling[30,31]. Similar to studies showing multiple distinct roles for LGP2, our data show that negative regulation is removed in LGP2$^{-/-}$ macrophages but also reveals other possible roles for LGP2 in governing immune polarization gene expression, supporting LGP2 as an RLR signaling cofactor[32,33] rather than an RLR itself, but additional studies will be required to fully distinguish the role of LGP2 in modulating the anti-WNV response in macrophages.

Our bioinformatics analysis identified ATF4 and SMAD4 as potential regulators in innate immune responses to viruses. By inhibiting these TFs, RIG-I and MDA5 may impart macrophage

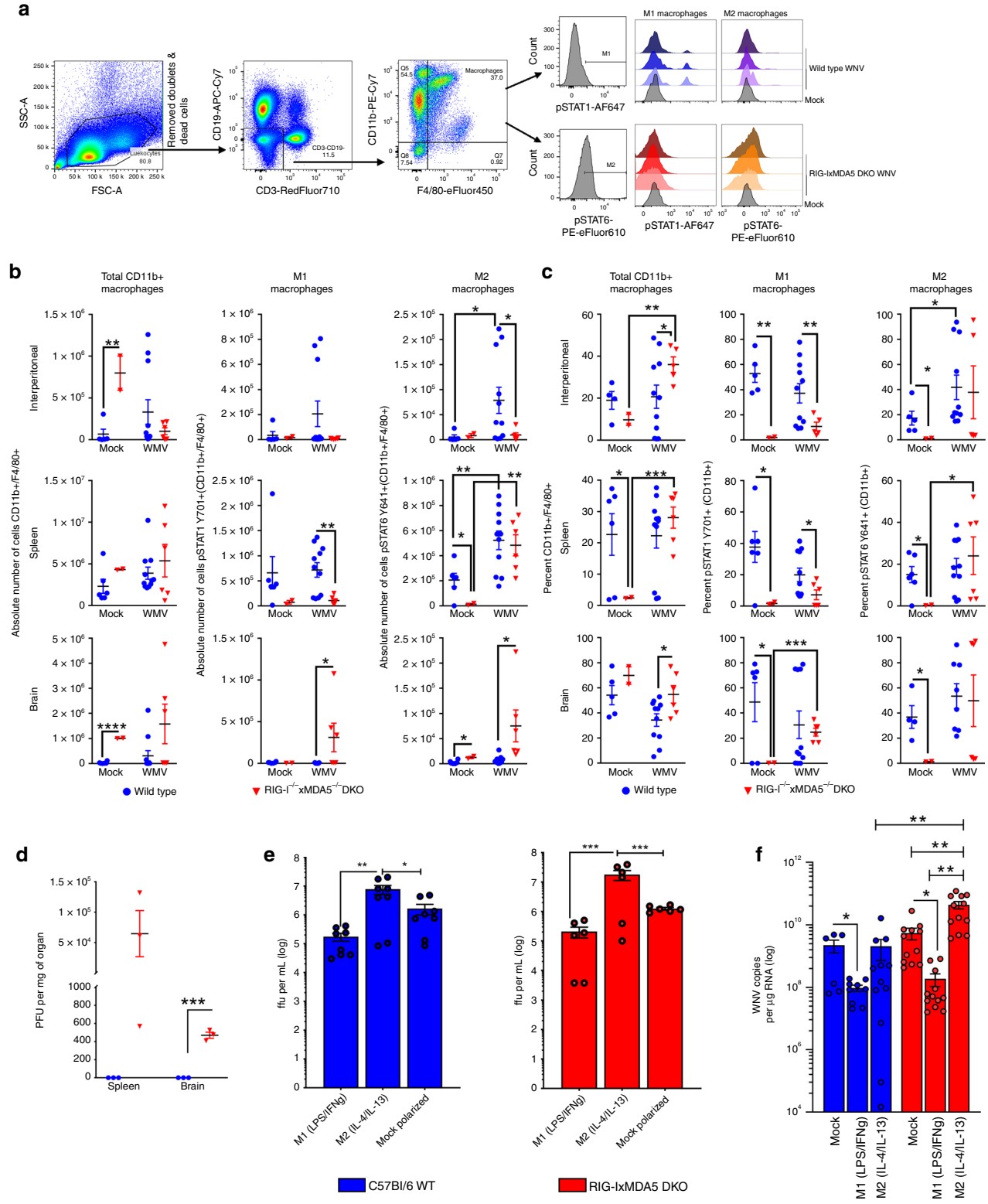

polarization and function. ATF4 is a known stress response factor that regulates gene expression to recruit macrophages to sites of inflammation[25,26], but here BMMs with intact RLRs suppress the ATF4 response, suggesting that a cell-level stress response is not productive in controlling WNV. SMAD4 has previously been reported to control BAFF production in macrophages[27], which contributes to downstream antibody responses through AID

expression in B cells[34]. Our data indicate that downregulation of SMAD4 may be RLR-dependent and may help to polarize the overall immune response towards an M1/Th1 cytotoxic response as opposed to an M2/Th2 antibody-based response. These two pathways may work in concert within the macrophage to mediate macrophage and immune polarization. Macrophage polarization has been directly linked to viral outcomes, and manipulation of

**Fig. 6** RIG-I and MDA5 are essential for macrophage polarization in vivo during WNV infection. WT or RIG-I/MDA5 DKO mice were infected with 100 pfu of WNV for 6 days. Spleens, brains, and IP lavage were harvested and analyzed. **a** Flow cytometry gating strategy for identification of CD11b[+] F4/80[+] cells, M1, and M2 populations. Representative histograms for STAT1pY701 and STAT6pY641 shown on right from three infected spleen samples and a representative mock from each genotype. **b** Absolute number of CD11b[+] F4/80[+] cells (left column), M1 (STAT1pY701[+]) cells (middle) and M2 (STAT6pY641[+]) cells (right column) from the IP lavage (top), spleen (center), or brain (bottom) from WT mice (blue) and RIG-I/MDA5 DKO (red). **c** Frequency of CD11b[+] F4/80[+] cells (left column), M1 (STAT1pY701[+]) cells (middle) and M2 (STAT6pY641[+]) cells (right column) as measured by flow cytometry from the IP lavage (top), spleen (center), or brain (bottom) from WT mice (blue) and RIG-I/MDA5 DKO (red). **d** PFU per mg of tissue for spleens and brains from the mice assayed in previous panels. Each symbol represents a single organ from WT mice (blue) or RIG-I/MDA5 DKO mice (red). The lines are the means ± SEM. $n = 3$ mice per group per genotype. **e** WT (blue) or RIG-I/MDA5 DKO (red) BMMs were in vitro polarized and infected with WNV. FFU per mL of supernatant are shown. **f** RNA from in vitro polarized, WNV-infected WT (blue) or RIG-I/MDA5 DKO (red) BMMs was assayed for genomic copies of WNV. For panels **e**, **f**, bars are means ± SEM. For panels **b**, **c**, **e**, **f**, data are combined from $n = 3$ independent experiments, and lines are the means ± SEM with the following total mouse numbers: Mock WT $n = 6$; Mock DKO $n = 2$; WNV WT $n = 11$; WNV DKO $n = 6$. Stars indicate statistical significance between the marked groups (Welch's unpaired $t$-test). $*p < 0.05$, $**p < 0.01$, $***p < 0.001$, $****p < 0.0001$. Source data are provided as a Source Data file

macrophage polarization can effectively alter viral disease progression[35]. Mouse microglia take on proinflammatory/M1 macrophage characteristics during WNV infection that protects neighboring cells from cytotoxic effects[36]. Our study connects these observations to peripheral macrophages and in vivo infection in which M1 polarization, potentially driven by RLR recognition of WNV, confers control of viral replication and neuroinvasion (Fig. 6). The generation of macrophage-specific RLR knockout mice will shed insight on the specific contribution of RLR programming in macrophages during WNV infection in vivo in future studies.

While TLR3 and 7 are expressed in macrophages, loss of RIG-I and MDA5 ablates transcriptional responses to WNV infection though TLRs are still expressed and functional. TLRs play an important role in WNV defense in vivo and in neurons[37] but our data show that in macrophages, the RLRs are the dominant sensing receptors. TLR3 has previously been shown to direct a response that facilitates neuroinvasion by WNV[38], supporting that RLR-based recognition leads to effective antiviral responses against WNV while TLR-based recognition does not. These disparate outcomes of TLR versus RLR immune protection have implications for vaccine adjuvant and antiviral therapeutic strategies that target innate immune factors for immune enhancement and flavivirus control. Targeting RLRs for antiviral actions could provide a polarized innate immune response for the control of WNV infection.

Our observations support a model where pathogen recognition and signaling by RLRs during acute WNV infection leads peripheral monocytes and tissue-resident macrophages into an M1 phenotype through parallel regulation of ATF4/SMAD4 with canonical STAT signaling to mediate M1 gene induction and M2 gene suppression. The resulting M1 macrophages would then directly restrict WNV replication and suppress virus spread. Effector functions of M1 macrophages may then facilitate the activation and polarization of the adaptive immune response to promote viral clearance. This outcome would depend on rapid signaling and sensing of WNV PAMPs wherein loss of RLRs compromise this response and increase susceptibility to neuroinvasion. Robust M1 macrophage activation by RLRs is critical for protection against WNV, and potentially for other flaviviruses.

## Methods
### Experimental model and subject details
Mice: All animal procedures were performed in compliance with all ethical regulations regarding animal-based research and this study received ethical approval from the University of Washington Institutional Animal Care and Use Committee. Mice (Mus musculus) were housed at the University of Washington SPF animal facility in Allentown filtered air cages with water bottles and no more than five mice of the same sex per cage. The mice used in these studies were healthy sex-matched females and males

between 8–10 weeks in age that were naïve to any previous experimentation. The C57Bl/6J (WT) mice were purchased from Jackson Laboratories. All other genotypes used were bred in-house. MDA5[−/−], and LGP2[−/−] have a 100% C57Bl/6J background, while the RIG-I[−/−], RIG-I[+/+] WT, and DKO all have a mixed C57Bl/6J and 129 × 1/SvJ background (F3 backcross from 129 × 1/SvJ to C57Bl/6). At the time of the experiments, mice weighed between 16.00–26.00 g.

The experimental groups (3–5 mice per group per experiment) were infected with 100 pfu WNV-TX in the footpad under anesthesia while the control groups were injected with an equal volume of PBS in the same procedure as the experimental groups. This subcutaneous footpad injection has been used previously to mimic the infection route of WNV through its mosquito vector[39]. Mice from each genotype were randomly assigned to either the experimental or control groups in equal numbers. Each mouse was monitored individually for clinical symptoms and weight loss over 6 (immune polarization experiments) or 21 days (morbidity and mortality experiments), euthanizing mice as they reached 20% initial weight loss or severe clinical symptoms. The number of mice used was determined via power calculation with an alpha of 0.05.

Spleens, brains, and IP lavage were collected from mice day six post-infection for analysis. Half of each organ (spleen and brain) was lysed using the Percellys lysis system and used for plaque forming unit (PFU) assay. Half of the spleens were processed for flow cytometer by disruption using the GentleMacs system (Miltenyi), then cells were stained as described below. Half of the brains were mashed through a 0.7-um tissue sieve (Bellco) and submitted to a hypertonic percoll gradient (1:10 10XPBS to Percoll [GE]) to remove excess myelin. Cells were then stained for flow cytometry.

Primary mouse cells: Femurs from C57Bl/6, MDA5[−/−], RIG-I[+/+], RIG-I[−/−], LGP2[−/−], and DKO mice (mixed male and female for each genotype) were harvested via sterile post-mortem extraction and bone marrow was isolated. After red blood cell lysis, the collected bone marrow was frozen and stored in liquid nitrogen. Upon defrosting, bone marrow cells were plated in DMEM + 10% FBS + 1X NEAA + 1X L-glutamine + 1X Na-pyruvate + 1 mM HEPES + 1X Gentamicin (cDMEM-10) with 40 ng per mL murine MCSF for 7 days, replenishing the media on day 3 and day 6 to differentiate the cells into macrophages (bone-marrow-derived macrophages [BMMs]).

### Method details
Virus: The WNV-TX02 infectious clone was isolated in the Gale Laboratory[18,40]. Stocks were generated for this study by infecting Vero cells at an MOI of 0.1 and harvesting filtered supernatant at day 5 post-infection.

Viral foci forming unit assays: Viral foci forming unit assay were performed on Vero cells in 96-well plates. Cells were infected for 2 h with dilutions of cellular supernatants. At that point, 125 uL of Methylcellulose medium was added to each well. The cells were then incubated for an additional 22 h at which point, the cells were fixed with 4% PFA in PBS. Plates were washed, and cells were stained with a FITC-conjugated anti-WNV E protein antibody (1:500 diltuion; kindly provided by Diamond Lab). After 2 h of staining, the cells were washed, and full wells were imaged on the Incucyte (Essen). FFU were calculated for each sample based upon the number of detected foci per well in triplicate wells.

Viral plaque forming unit assays: Viral plaque forming unit assays were performed on Vero cells in 6-well plates. Cells were infected for 2 h with dilutions of lysed organs in media. Then 2 mL of overlay (50% 1% agarose in water, 40% 2X DMEM, 5% NaHCO₃, 5% heat-inactivated FBS) was added to each well. The cells were then incubated for an additional 70 h at which point 2 mL of developing overlay (50% 1% agarose in water, 37% water, 10% 10X DPBS, 3% Neutral Red solution [Sigma]) was added. The neutral red overlay was allowed to develop for 6–8 h and then visible plaques were counted. PFU per mg of the organ were calculated for each sample based on the visible plaques per well in duplicate wells and the mg of tissue added to each well.

Infection of cells: BMMs were prepared as described above, then lifted, counted and plated at $1 \times 10^6$ cells per mL. Cells were allowed to adhere overnight and then infected with WNV-TX infectious clone at a multiplicity of infection (MOI) of

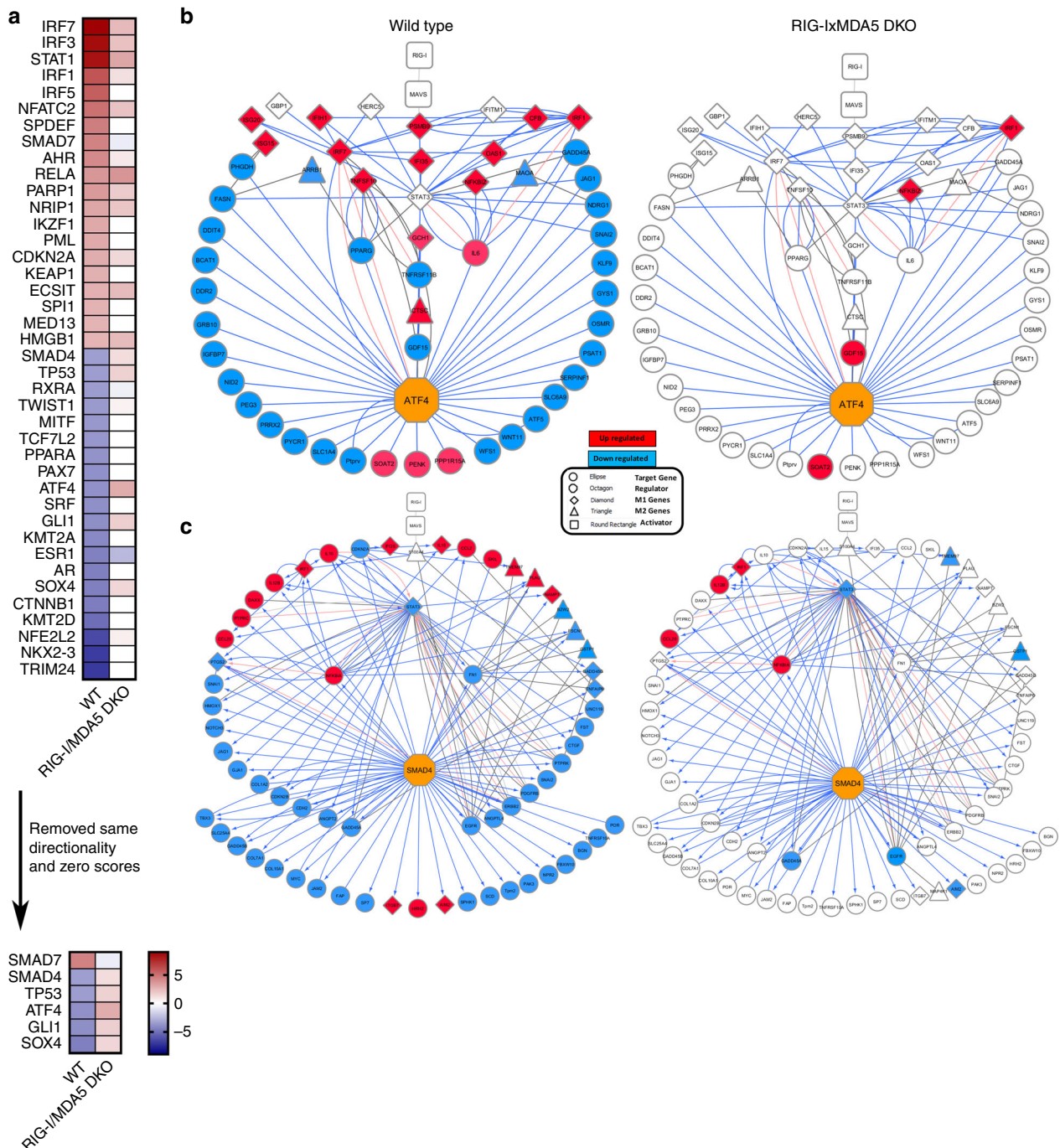

**Fig. 7** ATF4 and SMAD4 are potential mediators of the RLR-based macrophage response to WNV. **a** Upstream regulator analysis predicted regulators (Z-scores) are shown by heatmap where red shows predicted activation and blue shows predicted inhibition. This analysis was filtered to remove all regulators that had the same directionality in WT and DKO data sets as well as any regulator that had a zero Z-score in either dataset (bottom). **b**, **c** Network analysis of ATF4 target genes (**b**), SMAD4 target genes (**c**) and M1/M2 signature genes (both) in the WT response (left) and the DKO response (right). Predicted upstream transcriptional regulators are noted as orange octagons with the remaining nodes in our network are labeled by gene name. In the networks, shapes indicate gene category (circles—transcription factor target gene, octagon—regulator of interest, diamonds—M1 genes, triangles—M2 genes, and squares—activators). The colors are the directionality of the gene in our dataset (red is induced, blue is suppressed, and white is not changed compared to mock). $n = 3$ independent infections, RNA preparations, and sequencing results. Source data are provided as a Source Data file

2.5 for 2 h at which point the inoculum was removed and replaced with fresh media. Cells were then incubated for 22 or 46 additional hours. Cells were then harvested for RNA (Qiagen RNeasy Mini kit, 74106, as per manufacturer's instructions) or protein (described below) and the supernatants were harvested and stored at −80 °C.

Macrophage polarization: BMMs were prepared as described above, then lifted, counted, and plated at $1 \times 10^6$ cells per mL. Cells were allowed to adhere overnight and then stimulated with recombinant murine IFNγ (50 ng per mL final

concentration; Peprotech) and LPS (100 ng per mL final concentration; B5 Invivogen) for M1 polarization, recombinant murine IL-4 (100 ng per mL final concentration; Peprotech) and IL-13 (100 ng per mL final concentration; Shenandoah) for M2 polarization, or PBS for Mock polarization for 24 h. RNA, cell lysates for arginase activity (described below), and supernatants were harvested as described above. For infection control experiments after 24 h of polarization, the stimulating media was removed and infected as described above. After 22 h of additional incubation, cells were harvested for RNA and supernatants.

**Table 1 Reagents table**

| Reagent Type | Description | Source | Identifier |
|---|---|---|---|
| Antibody | Goat polyclonal anti-West Nile Virus NS3 (1:1000 dilution) | R&D systems | Cat#BAF2907; RRID: AB_2215927 |
| Antibody | Rabbit monoclonal anti-RIG-I (D14G6) (1:1000 dilution) | Cell Signaling Technologies | Cat#3743S; RRID: AB_2269233 |
| Antibody | Rabbit polyclonal anti-MDA5 (1:1000 dilution) | ProSci | Cat#4037; RRID: AB_735447 |
| Antibody | Rabbit polyclonal anti-LGP2 (1:1000 dilution) | Proteintech Group | Cat#11355-1-AP; RRID: AB_2092319 |
| Antibody | Goat polyclonal anti-Actin (I-19) (1:1000 dilution) | Santa Cruz Biotechnology, Inc | Cat#sc-1616; RRID: AB_630836 |
| Antibody | Rat anti-CD283 (TLR3) PE (1:100 dilution) | Biolegend | Cat#141903; lot#B183765; RRID: AB_10895749 |
| Antibody | Mouse anti-CD287 (TLR7) PE (clone: A94B10) (1:100 dilution) | BD Biosciences | Cat#565557 lot#5117683 |
| Antibody | Rat anti-CD11b PE-Cy7 (clone: M1/70) (1:160 dilution) | Thermo Fisher Scientific | Cat#25-01112-82; lot#E07514-1633; RRID: AB_469588 |
| Antibody | Rat anti-MHCII AlexaFluor700 (clone: M5/114.15.2) (3:1000 dilution) | Thermo Fisher Scientific | Cat#56-5321-82; lot#E09021-1631; RRID: AB_494009 |
| Antibody | Humanized E16 mouse anti-WNV E (1:500 dilution) | Michael Diamond's Lab[47] | N/A |
| Antibody | Rat anti-CD19 APC-Cy7 (clone: 6D5) (1:200 dilution) | Biolegend | Cat#115530; lot#B228154;RRID: AB_830707 |
| Antibody | Rat anti-CD3 redFluor710 (Clone: 17A2) (1:40 dilution) | Tonbo Biosciences | Cat#80-0032-U100; lot# C003202516803; RRID: AB_2621971 |
| Antibody | Rat anti-F4/80 eFluor450 (clone: BM8) (1:40 dilution) | Thermo Fisher Scientific | Cat#48-4801-80 lot#4278803; RRID: AB_1548756 |
| Antibody | Mouse anti-phospho-STAT1 (pY701) AlexaFluor647 (clone: 4a) (1:10 dilution) | BD Biosciences | Cat#612597; lot# 7080509; RRID: AB_399880 |
| Antibody | Mouse anti-phosoho-STAT6 (pY641) PE-eFluor610 (Clone: CHI2S4N) (1:20 dilution) | Thermo Fisher Scientific | Cat#61-9013-41; lot# 4329183; RRID: AB_2574673 |
| Antibody | Arm. Hamster anti-CD11c PerCP-Cy5.5 (N418) (1:80 dilution) | Thermo Fisher Scientific | Cat # 45-0114-80; Lot#4299461 |
| Antibody | Rat anti-MerTK SuperBright 600 (DS5MMER) (1:20 dilution) | Thermo Fisher Scientific | Cat#63-5751-82; lot#4345210 |
| Antibody | Rat anti-MerTK SuperBright 702 (DS5MMER) (1:20 dilution) | Thermo Fisher Scientific | Cat#67-5751-82; lot#1942740 |
| Antibody | Rat anti-CD206 BV605 (C068C2) (1:20 dilution) | Biolegend | Cat#141721; lot#B248321 |
| Antibody | Mouse anti-CD64 BV786 (X54-5/7.1) (1:40 dilution) | BD Biosciences | Cat#741024; lot#8053613 |
| Antibody | Rat anti-CD14 PE (Sa2-8) (1:40 dilution) | Thermo Fisher Scientific | Cat#12-0141-82; lot#4344716 |
| Antibody | Rat anti-CD45 SuperBright 645 (30-F11) (1:40 dilution) | Thermo Fisher Scientific | Cat#64-0451-82; lot#4339223 |
| Mouse | Mouse: Wild Type B6: C57Bl/6J | The Jackson Laboratory | JAX: 000664 |
| Mouse | Mouse: RIG-I WT: B6.129x1/SvJ.RIG-I$^{+/+}$ | Akira Lab;[48] | N/A |
| Mouse | Mouse: RIG-I$^{-/-}$:B6.129x1/SvJ.RIG-I$^{-/-}$ | Akira Lab;[48] | N/A |
| Mouse | Mouse: MDA5$^{-/-}$: B6.MDA5$^{-/-}$ | Colonna Lab;[5,49,50] | N/A |
| Mouse | Mouse: DKO: B6.129x1/SvJ | Gale Lab[5] | N/A |
| Mouse | Mouse: LGP2$^{-/-}$: B6.LGP2$^{-/-}$ | Gale Lab[17] | N/A |
| Primer | Mm *18S rRNA* primer | Qiagen | QT02448075 |
| Primer | Mm *IFNB* primer | Qiagen | QT00249662 |
| Primer | Mm *Rrpl37* primer | Qiagen | QT00112266 |
| Primer | Mm *IL-10* primer | Qiagen | QT00106169 |
| Primer | Mm *IL-1b* primer | Qiagen | QT01048355 |
| Primer | Mm *IL-6* primer | Qiagen | QT00098875 |
| Primer | Mm *NOS2* primer | Qiagen | QT00100275 |
| Primer | Mm *TNFa* primer | Qiagen | QT00104006 |
| Primer | Mm *CXCL1* primer | Qiagen | QT00115647 |
| Primer | Mm *Pparg* primer | Qiagen | QT00100296 |
| Primer | Mm *Chil3* primer | Qiagen | QT00108829 |
| Primer | Mm *Arg1* primer | Qiagen | QT00134288 |

RNA sequencing: RNA from cells infected as above were subjected to RNAseq (Expression Analysis, NC). The bioinformatics analysis of this data was performed in conjunction with The Center for Innate Immunity and Immune Disease (CIIID) Immune-informatics core group. The 48-hour DKO infection exhibited high cytopathic effects and thus reduced RNA quality to the extent that this sample point needed to be excluded from analysis.

RT-PCR: RNA from cells infected as above were reverse-transcribed into cDNA (Bio-Rad iScript or Qiagen QuantiTect RT kit, 205313, as per manufacturer's instructions). cDNA was analyzed by RT-PCR using listed primers purchased from Qiagen, SYBR Green master mix (QuantiFast SYBR Green PCR kit, 204057 or ThermoFisher SYBR Green PCR Master Mix, 4309155) were used as per manufacturer's instructions. PCR reactions were run on a 384-well Viia7 (Applied Biosystems) and analyzed using the ΔΔCT method.

WNV copy number RT-PCR: From cDNA generated as described above, WNV genomic copies were amplified and detected using TaqMan reagents (Forward 5' CCTGTGTGAGCTGACAAACTTAGT 3'; Reverse 5' GCGTTTTAGCATATTGACAGCC 3', Probe 5' 6FAM CCT GGT TTC TTA GAC ATC GAG ATC TTC TGT G C TAMRA 3', and TaqMan Virus Master Mix Life Technologies (#444432) then compared to a standard curve.

Western blots: Protein was isolated from infected cells by lysing the cells in a modified RIPA buffer (Tris-HCl 50 mM, 1% NP-40, 0.25% Na-deoxycholate, NaCl 150 mM, EDTA 1 mM pH 7.4) with protease and phosphatase inhibitors (Calbiochem 80501-130, Okadaic Acid [Fisher 49-560-4100UG], EDTA [100 uM], PMSF [200 uM] Sigma P8340). Cell lysates were agitated for a minimum of 30 min at 4 °C and then cleared via centrifugation. Cleared lysates were quantified by BCA assay (Thermo Scientific Fisher, PI-23221, and PI- 23224). Cell lysates were then separated by SDS-PAGE and transferred to a nitrocellulose membrane using a wet transfer system. Membranes were blocked with PBS- or TBS-based Odyssey Blocking buffer (Li-COR), washed, and proteins were analyzed by immunoblotting with standard methods using antibodies specific to listed proteins. Secondary antibodies conjugated to either AlexaFluor 680 or AlexaFluor 790 (1:10,000 dilution) were obtained from Jackson ImmunoResearch and Li-COR. Immunoreactive bands were detected on the Li-COR Odyssey Scanner. When required, blots were stripped with 0.2 M NaOH for five minutes.

Flow cytometry: Cells were prepared and, as needed, infected as described above. Cells were gently washed with DPBS (Ca$^{2+}$ and Mg$^{2+}$ free) and stained with V450 or V506 Fixable Live/Dead stain (eBioscience). Cells were then washed in FACS Wash (PBS with 0.016% sodium azide, 0.6% BSA), resuspended in FACS Wash containing fluorescently labeled antibodies, and incubated at 4 °C for 30 min. Cells were then washed twice in FACS Wash and resuspended in Brilliant Violet fluorescently labeled antibodies diluted in Brilliant Violet Staining Buffer (BD Biosciences) and incubated for 30 min at 4 °C. Cells were then washed twice in FACS wash and resuspended in 4% PFA in PBS for 20 min. Cells were then permeabilized with BD Perm/Wash Buffer (BD Biosciences) for 1 h 4 °C then pelleted and resuspended with intracellular antibodies in BD Perm/Wash Buffer for 30 min. After a final wash, cells were acquired on an LSRII flow cytometer (BD Biosciences). Flow cytometry data were analyzed using the FlowJo Software (TreeStar).

**Table 2 Statistical details for Fig. 6c WT vs RIG-I/MDA5 DKO: counts percentages**

| Organ | Graph | Cond. | P-value | T-value | DF |
|---|---|---|---|---|---|
| IP | CD11b+ | Mock | 0.0043 | 4.955 | 5 |
| IP | CD11b+ | WNV | 0.158 | 1.511 | 11 |
| IP | M1 | Mock | 0.607 | 0.5521 | 4 |
| IP | M1 | WNV | 0.080 | 1.952 | 10 |
| IP | M2 | Mock | 0.659 | 0.4796 | 4 |
| IP | M2 | WNV | 0.025 | 2.61 | 11 |
| Spleen | CD11b+ | Mock | 0.063 | 2.376 | 5 |
| Spleen | CD11b+ | WNV | 0.396 | 0.8739 | 15 |
| Spleen | M1 | Mock | 0.128 | 1.813 | 5 |
| Spleen | M1 | WNV | 0.002 | 4.103 | 11 |
| Spleen | M2 | Mock | 0.015 | 3.592 | 5 |
| Spleen | M2 | WNV | 0.729 | 0.354 | 12 |
| Brain | CD11b+ | Mock | <0.001 | 48.23 | 6 |
| Brain | CD11b+ | WNV | 0.172 | 1.56 | 6 |
| Brain | M1 | Mock | 0.083 | 2.134 | 5 |
| Brain | M1 | WNV | 0.0240 | 2.51 | 15 |
| Brain | M2 | Mock | 0.0158 | 3.33 | 6 |
| Brain | M2 | WNV | 0.0095 | 2.974 | 15 |
| IP | CD11b+ | Mock | 0.136 | 1.855 | 4 |
| IP | CD11b+ | WNV | 0.038 | 2.297 | 14 |
| IP | M1 | Mock | 0.002 | 7.363 | 4 |
| IP | M1 | WNV | 0.008 | 3.127 | 13 |
| IP | M2 | Mock | 0.041 | 2.962 | 4 |
| IP | M2 | WNV | 0.870 | 0.1715 | 6 |
| Spleen | CD11b+ | Mock | 0.028 | 3.058 | 5 |
| Spleen | CD11b+ | WNV | 0.282 | 0.1117 | 14 |
| Spleen | M1 | Mock | 0.014 | 3.672 | 5 |
| Spleen | M1 | WNV | 0.030 | 2.393 | 15 |
| Spleen | M2 | Mock | 0.012 | 3.83 | 5 |
| Spleen | M2 | WNV | 0.619 | 0.5205 | 7 |
| Brain | CD11b+ | Mock | 0.213 | 1.513 | 4 |
| Brain | CD11b+ | WNV | 0.037 | 2.391 | 10 |
| Brain | M1 | Mock | 0.026 | 3.136 | 5 |
| Brain | M1 | WNV | 0.616 | 0.515 | 12 |
| Brain | M2 | Mock | 0.029 | 3.925 | 3 |
| Brain | M2 | WNV | 0.874 | 0.1639 | 7 |

**Table 3 Statistical details for Fig. 6c: Mock vs. WNV (within genotype) counts percentages**

| Organ | Graph | Geno. | P-value | T-value | DF |
|---|---|---|---|---|---|
| IP | CD11b+ | WT | 0.119 | 1.67 | 13 |
| IP | CD11b+ | DKO | 0.170 | 3.344 | 1 |
| IP | M1 | WT | 0.134 | 1.61 | 12 |
| IP | M1 | DKO | 0.405 | 1.277 | 2 |
| IP | M2 | WT | 0.019 | 2.761 | 11 |
| IP | M2 | DKO | 0.842 | 0.2134 | 4 |
| Spleen | CD11b+ | WT | 0.182 | 1.417 | 12 |
| Spleen | CD11b+ | DKO | 0.610 | 0.543 | 5 |
| Spleen | M1 | WT | 0.877 | 0.1601 | 7 |
| Spleen | M1 | DKO | 0.530 | 0.7042 | 3 |
| Spleen | M2 | WT | 0.003 | 3.462 | 15 |
| Spleen | M2 | DKO | 0.002 | 5.587 | 5 |
| Brain | CD11b+ | WT | 0.205 | 1.353 | 10 |
| Brain | CD11b+ | DKO | 0.500 | 0.7271 | 5 |
| Brain | M1 | WT | 0.122 | 1.797 | 6 |
| Brain | M1 | DKO | 0.131 | 1.807 | 5 |
| Brain | M2 | WT | 0.178 | 1.416 | 15 |
| Brain | M2 | DKO | 0.105 | 1.975 | 5 |
| IP | CD11b+ | WT | 0.808 | 0.258 | 12 |
| IP | CD11b+ | DKO | 0.003 | 5.82 | 5 |
| IP | M1 | WT | 0.158 | 1.503 | 13 |
| IP | M1 | DKO | 0.063 | 2.518 | 4 |
| IP | M2 | WT | 0.046 | 2.19 | 14 |
| IP | M2 | DKO | 0.155 | 1.175 | 4 |
| Spleen | CD11b+ | WT | 0.961 | 0.0502 | 9 |
| Spleen | CD11b+ | DKO | <0.001 | 7.613 | 5 |
| Spleen | M1 | WT | 0.141 | 1.661 | 7 |
| Spleen | M1 | DKO | 0.142 | 1.712 | 6 |
| Spleen | M2 | WT | 0.487 | 0.7154 | 13 |
| Spleen | M2 | DKO | 0.046 | 2.647 | 5 |
| Brain | CD11b+ | WT | 0.064 | 2.165 | 8 |
| Brain | CD11b+ | DKO | 0.215 | 1.532 | 3 |
| Brain | M1 | WT | 0.363 | 0.9534 | 10 |
| Brain | M1 | DKO | <0.001 | 8 | 5 |
| Brain | M2 | WT | 0.247 | 1.238 | 9 |
| Brain | M2 | DKO | 0.065 | 2.361 | 5 |

Nitrate concentration analysis: Cellular supernatants were analyzed for concentration of nitrate using a Griess Assay kit (Promega) as per the manufacturer's instructions.

Arginase activity analysis: BMMs polarized as described above were lysed and analyzed for arginase activity using the QuantiChrom Arginase Assay kit (BioAssay Systems) as per the manufacturer's instructions.

For antibodies, mouse lines, and primers, see Table 1.

**Software availability**. All software tools (R/Bioconductor, star, cut adapt, ht-seq) are free and open source.

**Quantification and statistical analysis**. RNAseq data processing and analysis: Raw RNAseq data (FASTQs) were checked for quality (FastQC version 0.11.3), then adapters and rRNA were digitally removed (cut adapt, version 1.8.3 and Bowtie2 version 2.2.5). Roughly thirty million raw reads were mapped against the WNV viral and the mm10 (mouse) reference genomes separatel[41]. mm10 genome content was obtained from Illumina's igenomes site (https://support.illumina.com/sequencing/sequencing_software/igenome.html). Host alignments were performed using STAR (2.4.2) against UCSC MM10 and then converted into gene counts with ht-seq (0.6.0). Viral alignments were performed using Bowtie2 version 2.2.5 against WNV TX 2002-HC DQ17663[40]. Gene counts were then loaded in the R statistical programming language (version 3.2.0) and filtered by a mean of ten or greater across all samples.

A mm10 gene count matrix was then normalized using the voom package in R/Bioconductor. Differential expression was performed using the limma package in R/Bioconductor. Additional graphics packages were utilized for the visualization of numbers of differentially expressed genes (ggplot2), radial plots (plotrix), and heatmaps (gplots).

Co-expression heatmap: Co-expression was performed only on genes that were determined to be statistically significant from the differential expression analysis (threshold: log2 fold change $\geq |2|$ and FDR $\leq 0.05$) in at least one comparison.

Correlations (ward clustering and Euclidean distance) were run on the union of log2FC values using the WGCNA and heatmap.2 Bioconductor packages in R[42–44].

Matrisome heatmap: The global list of differentially expressed genes were compared against known genes associated with the Matrisome[20] . We then plotted those Matrisome-associated DE genes in a heatmap.

Please see the R markdown documents for additional information [http://stone.galelab.org].

**IPA pathway and regulator analysis**. A list of statistically significant, differentially expressed genes (threshold of significance of a $> |2|$-fold change over mock with a Benjamini-Hochberg-adjusted $p$-value $\leq 0.05$) were uploaded into Ingenuity Pathway Analysis (IPA) for core analysis to identify enrichment of biological pathways. IPA produced a list of known biological functions with an enrichment score ($-\log$ p-value) determining how significant those genes are to each function and an activation z-score that indicates the proposed activation of that pathway (activated or inhibited). The z-score is based on knowledge of expression changes (and functions) in the Ingenuity knowledge base[45].

Venn diagrams: The total DE gene list was filtered by specific functional groups (Th1, Innate, ISG, and M1) and then these lists were compared using venny an online Venn diagram tool[46].

Additional statistics: Additional statistics were performed using GraphPad Prism statistical package (versions 7.03 and 7.04).

Figure 1a: Survival Curve comparisons were Log-rank (Mantel-Cox) tests between the WT and RLR-KO survival curves. WT vs. RIG-I$^{-/-}$ $p = 0.0478$; WT vs. MDA5$^{-/-}$ $p = 0.0018$; WT vs. MAVS$^{-/-}$ $p < 0.0001$ Chi-square $= 53.12$; WT vs. RIG-I/MDA5 DKO $p < 0.0001$ Chi-square $= 51.85$; WT vs. LGP2$^{-/-}$ $p < 0.0001$ Chi-square $= 21.72$. figure 1d: FFU comparisons were Ordinary one-way ANOVA performed for each time point with Holm-Sidak's multiple-comparisons post-test (two-tailed; 24h $t = 3.298$, DF $= 23$; 48h $t = 3.475$, DF $= 42$).

Figure 4b: Comparisons were two-tailed unpaired Holm-Sidak method with alpha $= 0.5$ DF $= 6$ (modified $t$-tests, p-values reported) $IL$-$6$ $p = 0.003183$ $t = 4.742$; $NOS2$ $p = 0.11348$ $t = 2.710$; $TNFa$ $p = 0.021581$ $t = 3.083$; $CXCL1$ $p = 0.124645$

$t = 1.784$; *PPARg* $p = 0.296637$ $t = 1.143$; *Chil3* $p = 0.991537$ $t = 0.01106$; *Arg1* $p = 0.242478$ $t = 1.296$. Figure 4c: MFIs were normalized to Mock polarized MFI values (arbitrarily set to 100) and comparisons were two-tailed unpaired Holm-Sidak method with alpha = 0.5 DF = 2 (modified *t*-tests, adjusted p-values reported). MerTK – M1 $p = 0.8267$ $t = 0.4631$, M2 $p = 0.8267$ $t = 0.6476$; CD64 - M1 $p = 0.3676$ $t = 1.855$, M2 $p = 0.4142$ $t = 1.022$; CD14 - M1 $p = 0.0307$ $t = 7.943$, M2 $p = 0.4361$ $t = 0.9656$; CD11c - M1 $p = 0.7118$ $t = 0.9$, M2 $p = 0.7118$ $t = 0.4311$; CD206 - M1 $p = 0.5029$ $t = 0.8102$, M2 $p = 0.4357$ $t = 1.61$. Figure 4d, g: Comparisons were two-tailed unpaired Holm-Sidak method with alpha = 0.5 DF = 16 (modified *t*-tests, adjusted p-values reported) *IL-10* $p = 0.9229$ $t = 0.5117$; *IL-1b* $p = 0.9229$ $t = 0.7112$; *IL-6* $p = 0.1028$ $t = 2.638$; *NOS2* $p = 0.9229$ $t = 0.2915$; *TNFa* $p = 0.9229$ $t = 0.7349$; *CXCL1* $p = 0.7717$ $t = 1.179$; $t = 1.092$; *PPARg* $p = 0.6401$ $t = 0.4766$; *Chil3* $p = 0.3462$ $t = 1.587$; *Arg1* $p = 0.2594$ $t = 1.924$. Figure 4e: Comparisons were two-tailed unpaired Holm-Sidak method with alpha = 0.5 (modified *t*-tests, adjusted p-values reported) M1 $p = 0.1872$ $t = 1.709$ DF=28; Mock $p = 0.8232$ $t = 0.2262$ DF=22. Figure 4f: Comparisons were two-tailed unpaired Holm-Sidak method with alpha = 0.5 DF = 4 (modified *t*-tests, adjusted p-values reported) M2 $p = 0.6322$ $t = 0.9552$; Mock $p = 0.7538$ $t = 0.3359$.

Figure 6b, c: percentage and count comparisons were two-tailed unpaired Welch's *t*-test between the genotypes for each organ and infection condition (Tables 2, 3).

Figure 6d: PFU comparisons were one tailed *t*-tests between genotypes for each organ. Spleen $p = 0.2309$ $t = 1.702$ DF = 2; Brain $p = 0.0052$ $t = 13.88$ DF=2. Figure 6e Comparisons were two-tailed unpaired Holm-Sidak method with alpha = 0.5 (modified *t*-tests, adjusted p-values reported) WT – DF = 21 M1 vs M2 $p = 0.007$ $q = 4.833$, M1 vs Mock $p = 0.7874$ $q = 0.937$, M2 vs Mock $p = 0.0306$ $q = 3.896$; RIG-I/MDA5 DKO DF = 36 M1 vs M2 $p = 0.0003$ $q = 6.252$, M1 vs Mock $p = 0.9636$ $q = 0.3668$, M1 vs M2 $p = 0.0005$ $q = 5.885$. Figure 6f: Comparisons were unpaired two-tailed *t*-tests. WT – M1 vs Mock $p = 0.017$ $t = 2.744$ DF=13, M1 vs M2 $p = 0.225$ $t = 1.255$ DF=19, M2 vs Mock $p = 0.9288$ $t = 0.0907$ DF=16; RIG-I/MDA5 DKO M1 vs Mock $p = 0.037$ $t = 2.371$ DF=11, M1 vs M2 $p = 0.004$ $t = 3.67$ $t = 11$, M2 vs Mock $p = 0.008$ $t = 3.165$ DF=12; WT vs RIG-I/MDA5 DKO – Mock $p = 0.3369$ $t = 0,9901$ DF=16, M1 $p = 0.3536$ $t = 0.9509$ DF=19, M2 $p = 0.0021$ $t = 3.489$ DF=22.

**Reporting summary**. Further information on research design is available in the Nature Research Reporting Summary linked to this article.

## Data availability

Sequencing data are in Gene Expression Omnibus (GEO) under accession GSE104817. Data underlying Figs. 1A, B, D, 3D, 4B–G, 5B, 6B, C, E, F, 7A and Supp Figs. 1, 2A, 7E, 8A, 8B, and 10B are provided as Source Data files. All other data are available from the corresponding author upon reasonable requests.

## Code availability

R markdown data reports were generated for data analysis and figure reproducibility. They can be found here: [http://stone.galelab.org/]

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

## Acknowledgements

The authors would like to thank: Aimee Sekine and Kathleen Voss for their help with the mouse colony and mouse harvests; Connor Driscoll for bioinformatics support; Albert Huang for contributing to the ISG list; and our funding sources. The work was supported by grants AI104002, AI100625, and AI083019. A.E.L.S. was supported by T32 AI007509 and F32 AI124520-02. E.A.H. is supported by AHA award 17POST33660907.

## Author contributions

A.E.L.S.—designed, conducted and analyzed the experiments, and wrote the manuscript. E.A.H.—conducted and analyzed the experiments, and edited the manuscript. R.G.—analyzed experiments and edited the manuscript. C.W.—designed and analyzed experiments, and edited the manuscript. M.J.G.—designed and analyzed experiments, and edited the manuscript

## Additional information

**Competing interests:** The authors declare no competing interests.

