## [Peer Review File · Nature Communications]

Reviewers' comments:

Reviewer #1 (Remarks to the Author):

In this well written and thoughtfully-executed paper, the Gale group takes a deep dive to address the exact roles for the RLR system in response to WNV infections. This paper examines the RLR-activated gene expression profiles in primary BMM from WT and KO mice. Results show that RIG-I and MDA5 are the most important RLRs for WNV infections and that their presence correlates with M1 macrophage polarization in response to WNV and suppression of the M2 response. An informatics and computational analysis pipeline using their high quality datasets identifies ATF4 and SMAD4 as previously unappreciated transcription factors that mediated RLR-initiated macrophage polarization during WNV infections. These findings are competently demonstrated, reasonably interpreted, and provide detailed new information on the roles of RLRs in virus infections and new research directions for future study. Only minor questions/comments are raised primarily for my own education and may help clarify aspects of the paper:

- 1- Initial reports for some RLR KOs indicated some embryonic or perinatal morbidity- is this the case with any of the mice in the current study?
- 2- While the data support a role for ATF4 and SMAD4, more details about the quantitation of blots would be appreciated. The ATF4 signal shown is weak, and the effect on total SMAD4 in WT is unexplained. Please also comment on the mechanisms for SMAD degradation by WNV.
- 3- The effect of LGP2 KO in figure 1A is dramatic compared to MDA5 and LGP2, and this was explained by the prior discovery of T cell defects by this group. However it is likely that LGP2 has additional roles that could also be revealed in the context of RIG-I or MDA5 deficiency.

Reviewer #2 (Remarks to the Author):

This is a very nice and rich study focused on the role of RLRs in the response to macrophages against WNV infection. The generated information, even though mainly descriptive, is very interesting and of interest to the field, due to the large amount of data and the intriguing possible implications in macrophage biology. More caution however should be given to the discussion of the impact of these signaling pathways in macrophages in infections *in vivo*, as one would need macrophage specific kos to ascertain that the impact *in vivo* is due to RLR signaling in macrophages. This should be better discussed.

Reviewer #3 (Remarks to the Author):

The manuscript entitled "RIG-I-like receptors direct inflammatory macrophage polarization against West Nile virus infection" is a detailed report of the macrophage response to WNV infection in mice and its dependence on RIG-I-like receptors (RLRs). Previous data have shown that RLRs RIG-I and MDA5 are non-redundant in controlling WNV infection in mice and lead to low levels of innate immune signaling. The studies here extend those results, and they show that RLRs also control macrophage polarization in response to WNV infection using *ex vivo* studies. The overall transcriptional response to WNV infection in cells lacking RLRs is described in detail, though the effect of RLR knockout on innate immune responses to WNV infection has previously been established. The non-ISG response to WNV infection in RLR knockout cells is also described, and this includes macrophage polarization.

Specific comments

1. While the data suggests that ATF4 and SMAD4 are transcription factors associated with the RLR-dependent WNV-induced polarization of macrophages, the data thus far only shows linkage, and further studies would need to be performed to show that these transcription factors are responsible for regulating polarization through RLRs.
2. To test whether the lack of M1 polarization in cells from mice with the double knockout is

responsible for the increased pathogenesis in DKO mice in response to WNV infection, M1 polarization could be stimulated in DKO mice through classic stimuli.

We thank the reviewers for their kind comments and suggestions. We have responded to these comments below and in the editing of the manuscript. Please see the point by point response to the reviewers' comments below following each reviewer question or comment below in red font.

Reviewers' comments:

Reviewer #1 (Remarks to the Author):

In this well written and thoughtfully-executed paper, the Gale group takes a deep dive to address the exact roles for the RLR system in response to WNV infections. This paper examines the RLR-activated gene expression profiles in primary BMM from WT and KO mice. Results show that RIG-I and MDA5 are the most important RLRs for WNV infections and that their presence correlates with M1 macrophage polarization in response to WNV and suppression of the M2 response. An informatics and computational analysis pipeline using their high quality datasets identifies ATF4 and SMAD4 as previously unappreciated transcription factors that mediated RLR-initiated macrophage polarization during WNV infections. These findings are competently demonstrated, reasonably interpreted, and provide detailed new information on the roles of RLRs in virus infections and new research directions for future study. Only minor questions/comments are raised primarily for my own education and may help clarify aspects of the paper:

1- Initial reports for some RLR KOs indicated some embryonic or perinatal morbidity- is this the case with any of the mice in the current study?

While the RIG-I^{-/-} mice on a C57Bl/6 background do demonstrate embryonic morbidity, cross-breeding of these mice onto the A129 background has alleviated this issue. To ensure this cross-breeding did not alter the interpretation of our results, we compared RIG-I^{-/-} to RIG-I WT on the same mixed background. We did not observe significant differences between C57Bl/6 WT and RIG-I WT (Figure S2). In this cohort, we did not experience any embryonic lethality with the pure C57Bl/6 background mice (MDA5^{-/-}, LGP2^{-/-}) or the mixed background of C57Bl/6 and A129 (RIG-I^{-/-} and RIG-IxMDA5 DKO).

2- While the data support a role for ATF4 and SMAD4, more details about the quantitation of blots would be appreciated. The ATF4 signal shown is weak, and the effect on total SMAD4 in WT is unexplained. Please also comment on the mechanisms for SMAD degradation by WNV.

The data shown in those blots have been removed. Further analysis with SMAD4/7 and ATF4 is currently ongoing as part of a larger, follow up study and not included in this manuscript as the time to generate appropriate conditional knockout mice to test the intrinsic role of ATF4 and SMAD4 in RLR-dependent macrophage polarization can not be generated in the time allowed for resubmission. Therefore, we have revised the language in our manuscript to describe the potential linkage of ATF4 and SMAD4 to RLR-dependent polarization of macrophages based upon our RNAseq data.

3- The effect of LGP2 KO in figure 1A is dramatic compared to MDA5 and LGP2, and this was explained by the prior discovery of T cell defects by this group. However it is likely that LGP2 has additional roles that could also be revealed in the context of RIG-I or MDA5 deficiency.

The reviewer makes an excellent point here. The modulation of the signaling members of the RLR family by LGP2 remains unclear due to contradicting results in the literature. Our studies here did not include a double knockout of RIG-I and LGP2 or MDA5 and LGP2. These mice are currently of interest to our group and are in the process of being generated. However, they are not currently available to add to this study.

Reviewer #2 (Remarks to the Author):

This is a very nice and rich study focused on the role of RLRs in the response to macrophages against WNV infection. The generated information, even though mainly descriptive, is very interesting and of interest to the field, due to the large amount of data and the intriguing possible implications in macrophage biology. More caution however should be given to the discussion of the impact of these signaling pathways in macrophages in infections in vivo, as one would need macrophage specific kos to ascertain that the impact in vivo is due to RLR signaling in macrophages. This should be better discussed.

This point has been added to the discussion and the language has been softened to reflect this question.

Reviewer #3 (Remarks to the Author):

The manuscript entitled “RIG-I-like receptors direct inflammatory macrophage polarization against West Nile virus infection” is a detailed report of the macrophage response to WNV infection in mice and its dependence on RIG-I-like receptors (RLRs). Previous data have shown that RLRs RIG-I and MDA5 are non-redundant in controlling WNV infection in mice and lead to low levels of innate immune signaling. The studies here extend those results, and they show that RLRs also control macrophage polarization in response to WNV infection using ex vivo studies. The overall transcriptional response to WNV infection in cells lacking RLRs is described in detail, though the effect of RLR knockout on innate immune responses to WNV infection has previously been established. The non-ISG response to WNV infection in RLR knockout cells is also described, and this includes macrophage polarization.

Specific comments

1. While the data suggests that ATF4 and SMAD4 are transcription factors associated with the RLR-dependent WNV-induced polarization of macrophages, the data thus far only shows linkage, and further studies would need to be performed to show that these transcription factors are responsible for regulating polarization through RLRs.

The language surrounding the ATF4/SMAD4 data has been softened to describe the potential linkage of ATF4 and SMAD4 to RLR-dependent polarization of macrophages based upon our RNAseq data, and we have removed the ATF4 and SMAD4 protein analysis from

this study. Generation of conditional knockouts of ATF4 and SMAD4 in macrophages would not allow for the completion of these studies within the resubmission timeline. Therefore, further studies to determine the role of ATF4 and SMAD4 in RLR-dependent WNV-induced polarization of macrophages are ongoing as a follow up study, but are no longer included in this manuscript.

2. To test whether the lack of M1 polarization in cells from mice with the double knockout is responsible for the increased pathogenesis in DKO mice in response to WNV infection, M1 polarization could be stimulated in DKO mice through classic stimuli.

The reviewer makes an interesting point here. This experiment is currently outside of the scope of the current study. Future studies are planned to further examine the role of polarized macrophages in the control of flaviviruses *in vivo*.

REVIEWERS' COMMENTS:

Reviewer #1 (Remarks to the Author):

The authors have gone above and beyond to support their data with additional experiments and modify the language to better reflect the conclusions of their work.

Reviewer #2 (Remarks to the Author):

The authors have responded well to the minor recommendations of this reviewer. This study is a very nice addition to the study of viral innate immunity

Reviewer #3 (Remarks to the Author):

The authors have adequately addressed the reviewers' concerns.

The referees had no comments that required a response.

REVIEWERS' COMMENTS:

Reviewer #1 (Remarks to the Author):

The authors have gone above and beyond to support their data with additional experiments and modify the language to better reflect the conclusions of their work.

Reviewer #2 (Remarks to the Author):

The authors have responded well to the minor recommendations of this reviewer. This study is a very nice addition to the study of viral innate immunity

Reviewer #3 (Remarks to the Author):

The authors have adequately addressed the reviewers' concerns.